# Review of Organizational Health Literacy Practice at Health Care Centers: Outcomes, Barriers and Facilitators

**DOI:** 10.3390/ijerph17207544

**Published:** 2020-10-16

**Authors:** Elham Charoghchian Khorasani, Seyedeh Belin Tavakoly Sany, Hadi Tehrani, Hassan Doosti, Nooshin Peyman

**Affiliations:** 1Health Education and Promotion, Student Research Committee, Mashhad University of Medical Sciences, 13131–99137 Mashhad, Iran; khorasanie961@mums.ac.ir; 2Department of Health Education and Health Promotion, Faculty of Health, Mashhad University of Medical Sciences, 13131–99137 Mashhad, Iran; TavakkoliSaniB@mums.ac.ir (S.B.T.S.); TehraniH@mums.ac.ir (H.T.); 3Department of Mathematics and Statistics, Macquarie University, 2109 Sydney, Australia; hassan.doosti@mq.edu.au; 4Social Determinants of Health Research Center, Mashhad University of Medical Sciences, 13131–99137 Mashhad, Iran

**Keywords:** organizational health literacy, health literacy, health outcome, quality improvement, barriers, public health, health-literate healthcare service

## Abstract

The term organizational health literacy (OHL) is a new concept that emerged to address the challenge of predominantly in patients with limited health literacy (HL). There is no consensus on how OHL can improve HL activities and health outcomes in healthcare organizations. In this study, a systematic review of the literature was conducted to understand the evidence for the effectiveness of OHL and its health outcome, and the facilitators and barriers that influence the implementation of OHL. A literature search was done using six databases, the gray literature method and reference hand searches. Thirteen potentially articles with data on 1254 health organizations were included. Eight self-assessment tools and ten OHL attributes have been identified. Eleven quality-improvement characteristics and 15 key barriers were reviewed. Evidence on the effectiveness of HL tools provides best practices and recommendations to enhance OHL capacities. Results indicated that shifting to a comprehensive OHL would likely be a complex process because HL is not usually integrated into the healthcare organization’s vision and strategic planning. Further development of OHL requires radical, simultaneous, and multiple changes. Thus, there is a need for the healthcare system to consider HL as an organizational priority, that is, be responsive.

## 1. Introduction

In the late modern “health” and “multi-option” communities, health literacy is a specific skill that is needed to successfully deal with a large number of health relevant-tasks and decisions to be taken every day [1,2].

The concept of health literacy (HL) is used in the United States (mainly in medical centers) to study the effectiveness of HL on treatment outcome, especially in patients with insufficient and inadequate HL severity [2,3]. In this framework, there is evidence that HL must be regarded as a context or relationship strategy related to personal HL, but the needs of the organization for users must also be considered [4,5]. The concept of HL not only led to methods for estimating personal HL, but also to the estimation of the organization’s HL sensitivity [6,7]. Several health strategies are designed to improve health behavior, self-care abilities, and the quality of life, but adherence and participation are related to social health determinants [8,9]. A large study showed that HL was a main modifying factor for health outcomes because it is associated with social health determinants [10,11]. Although organizational initiatives responding to HL may be a practical approach to improve healthcare systems and health-promoting hospitals, using experiences from the quality movement in these systems remains less responsive than needed to the concept of limited HL [12,13]. Unintentional non-adherence to medications and treatments; lack of involvement of patients in decision making; difficulties with patients- provider communication, discharge instructions and informed consent; and increasing rates of hospitalizations, readmissions, and emergency care have been reported for patients with limited HL [14,15].

There is now a growing recognition that HL depends not only on individual skills and abilities but also on the demands and complexities of the health-care system [9,10]. One way to raise almost all low levels of HL in populations is to build health-literate systems of care. The term organizational health literacy (OHL) is a relatively new concept that emerged to address the challenge of most in patients with limited HL [9]. It describes a health care organization that uses strategies to make it easier for patients to engage in the health care process, navigate the health care system, understand health information, and manage their health [16,17,18]. Thus, healthcare quality could be improved and patients’ may be better met if healthcare organizations transformed into HL responsive ones delivering care in a way that protects HL skills and practices [9,10].

With a prima facie evidence, there has been a steady rise in designs and structures of OHL in the recent years [6,14], for example, the ten attributes of an OHL were defined by the Institute of Medicine (IOM) Health Literacy Roundtable, which is a first systematic attempt to implement the HL concept in health literate health care organizations. The attributes of OHL provide a guide to altering health care organizations to “make it easier for people to navigate, understand, and use information and services to take care of their health” [1,19,20]. These ten attributes focus on delivering health information, staff training in health communication, address HL with specific leadership activities and assess the quality of the organization’s environment for patients with different levels of HL. According to these attributes, an OHL: (1) has leadership that makes HL integral to its mission, structure, and operations; (2) integrates HL into planning, evaluation measures, patient safety and quality improvement; (3) prepare the workforce to be health literate and monitors progress; (4) include populations served in the design, implementation, and evaluation of health information and services; (5) meet the needs of people with various HL skills while avoiding stigmatization; (6) use HL strategies in interpersonal communication and confirm the understanding of all points of contact; (7) provide easy access to health information and services and navigation assistance; (8) design and distributes print, audiovisual, and social media content that is easy to understand and act on; (9) addresses health literacy in high-risk situations, including care transitions and communications about medicines; and (10) communicates clearly what health plans cover and what individuals will have to pay for services [21,22]. Likewise, a series of HL guides and operational frameworks have been developed in the past few years to help organizations with their transition to OHL [19,20].

Despite the recent research on designs and structures of OHL, the effectiveness and use of these attributes have not been reviewed systematically to understand the outcomes and implications of the transformation to the health-literate organization [23,24,25,26]. The scientific studies have focused most of its attention on patient treatment; however, the role of OHL in empowering patient’s self- management support and communication has been overlooked [27]. Likewise, limited work has been done to identify barriers and facilitators that influence quality improvement in OHL [9,28]. Absent such information, health systems may be unable to evaluate whether OHL-related attributes and guidelines have been implemented effectively or to identify the characteristics and outcomes of its environment most in need of improvement [29]. Therefore, in this review, we aim to address these specific gaps in knowledge by focusing on the following: (1) the evidence for the effectiveness of OHL practices and outcome at a healthcare center by using the criteria found in the 10 attributes, and (2) the facilitators and barriers that influence the implementation OHL.

We used a systematic review as a tool to conduct a comprehensive assessment of OHL improvements. This review provides insights into the implementation of OHL that are less visible to health providers and patients, and identify facilitators and barriers that influence the quality of the organization and delivery of healthcare systems. These findings have relevance for health providers, professionals, and researchers working to improve OHL and evaluating the quality of care.

## 2. Materials and Methods

We planned a systematic review based on the preferred reporting items for systematic reviews (PRISMA) [30] and Cochrane collaboration tool [31]. The main research questions were:Which tools are used for the assessment of OHL practice?How do OHL practices interact to improve organizational quality, thereby promoting sustainable HL activities and health outcomes?What factors facilitate the implementation of OHL?Which barriers interact to reduce or increase the quality of OHL?

### 2.1. Search Methods

Literature search in Cochrane database, PubMed, Scopus, the web of knowledge, MEDLINE, and Google scholar were performed without restriction in study type from inception until November 2019. The database search was supplemented by gray literature and reference hand searches. We used the Medical subject heading (MeSH) thesaurus and eight keywords (organization, organizational, healthcare center, health care services, primary care, patient-centered care, health facilities) combined with Boolean operators with the term health literacy. There are no study design and country restrictions. To increase the accuracy of the study, all scientific publications in full-text format included in this review. However, original research studies were given preference because they provide details about methods.

### 2.2. Data Abstraction and Article Screening

Studies were included for inclusion criteria if they related to OHL. Details of the exclusion and inclusion criteria were provided in Table 1. In this review, first, titles and abstracts of all studies were screened by two independent authors to select potentially eligible studies were downloaded in full text. In the second stage, results and methodology sections of full text reports were screened to determine eligibility. Both authors were in agreement overall studies included; and the third reviewer resolved any discrepancies and doubts and regarding the inclusion criteria.

### 2.3. Data Extraction and Quality Appraisal

We performed content analysis adapted the systematic review to conduct independent double data extraction and the following characteristics from all included studies was extracted: (1) studies year/author(s), country, study design and type, and aims; (2) health sector; (3) characteristics of OHL; (4) health outcome and practical implications of OHL; (5) checklists, guide and attributes, as it relates to the concept of OHL; and (6) barriers and facilitators to implementing OHL. All doubts and disagreements about data extraction were resolved through discussions between the authors, and an independent dual rating was also conducted based on the Cochrane diagnostic test accuracy review to assess the quality of the included studies [32]. The quality of the included studies was acceptable, with an average score of 11.5. The range is from 8.4 to 14. This is consistent with the QUADAS (quality assessment of diagnostic accuracy studies) guideline used in systematic review studies [33].

## 3. Results

### 3.1. Search Outcome and Study Design

The database search, combined with gray literature and reference tracking, identified 531 publications from 2006 to 2019: 235 from Google scholar, 117 from Scopus, 58 from PubMed, 121 from the web of science. Based on the inclusion criteria, 13 potentially relevant articles with data on 1254 health organizations (9 healthcare centers, 96 hospitals, 60 Pharmacies, 629 health clinics, and 453 cancer centers) were included in the systematic review (Figure 1, Table 2). We collected all data from 7 different countries, and 57% of studies were undertaken in the United States of America (USA) [34,35,36,37]. The first studies on OHL were conducted in 2006 [38] and others included articles that were published after 2012. We included 3 pilot studies [27,39,40], 3 cross-sectional studies [9,41,42], 2 case studies [34,37], 2 preliminaries [43,44], a comparative research piece [45], a workshop review [46], and a qualitative study [47]. The development and aim of 8 out of 13 studies (61.5%) [9,27,39,40,43,45,46,47] specifically focused on investigating HL issues in health organizations from the health provider—patients’ perspective and addressed the implementation of OHL, other studies (5 studies, 48.5%) [34,37,41,42,44] targeted the development of the assessment tool to measure OHL practice in health organizations (Table 2).

### 3.2. Assessment Tools for Organizational Health Literacy Practice

Eight assessment tools have been identified (Table 3). Assessment tools were developed and designed in their framework (6 to 10 OHL attributes) to assess OHL practices and activities at the individual and system levels. In this review, 6 (46%) studies [9,37,39,40,41,45] assessed OHL practices using the organizational health literacy observation (OHLO) [9], health literate health care organization 10 item questionnaire (HLHO-10) [9,37,40,41], health literacy-sensitivity of communication (HL-COM) [39], and Agency for Healthcare Research and Quality (AHRQ) [45] that refer to ten OHL attributes to address HL. Four studies (31%) [27,42,43,47] assessed HL issues using Vienna Health Literate Organization (VHLO) [27,47] and communication climate assessment toolkits (C-CAT) [42,43] tools that examine 9 OHL attributes. Three studies (23%) [34,44,46] assessed OHL practices using organizational health literacy responsiveness (Org-HLR) [46] and health literacy environment of hospitals and health centers (HLE) [34,44] that include 7 and 6 OHL attributes, respectively (Table 3). The majority of tools (6 tools) [9,37,43,44,46,47] have been developed for healthcare centers/clinics and Hospitals in general; one is specialized for pharmacies [45], and another one is designed to support health-literate cancer patient practices in cancer centers [39]. The findings showed that HLHO-10 and VHLO were the most common tools used to assess OHL.

### 3.3. Quality Improvement Characteristics of OHL Practice

Eleven quality improvement elements are found in the reviewed studies (Table 4). The following four elements—HL is an organizational priority (8 articles from 13 articles (8/13); 61%) [9,27,34,39,42,43,45,46], communication practices and standards (9/13; 69%) [9,27,34,37,39,42,44,45,47], verbal/written communication skills (8/13; 61%) [9,27,34,37,39,42,44,45], and patient knowledge and engagement (4/13; 31%) [34,39,43,47], are frequently considered as main characteristics to improve quality of OHL practices and health outcome. Two studies identified access to services and procedures (how well service support ongoing access and initial entry to service and programs and how services provide outreach services) [9,40], and continuity and integration of HL practices and care [27,37], navigation quality (how well service addresses strategies to simplify navigation of the health care system and healthcare facilities) [40,44], and workforce (how well service provides supportive working environments, practice tools and resources, and ongoing professional development) for improvement plans [37,44]. Some studies also included leadership and commitment (1/13; 7.8%) [37], HL friendliness of environments (1/13; 7.8%) [47], and financial management (1/13; 7.8%) [9].

Thirteen articles published in 2006-2019 described the application of the assessment tools for OHL practice [9,27,34,37,39,40,41,42,43,44,45,46,47]. The majority of these studies (8/13, 61.5%) [9,39,40,41,42,43,44,45] described the use of HL issues in health organizations from the health provider patients’ perspective; few articles detailed implementation of OHL (2/13, 15%) [34,37]. These studies allow us to assess evidence of the effects of OHL, they indicated that implementation of OHL tools in health organizations can adopt specific health-literate practices facilitating action to remedy HL barriers and to understand the complexity of treatment and the factors influencing diagnosis and treatment. Although studies on the implementation of OHL are scarce, existing literature confirm that systemic research to addressing HL within healthcare organizations is essential.

### 3.4. Barriers and Facilitators of OHL

In this review, 15 key barriers were identified (Table 5). These barriers are also conceived as facilitators that covering 5 broad themes: barriers 1 to 5 define organizational leadership and commitment; 6 to 8 refer to patient-providers communication skills and practices; 8 to 12 relate to patient knowledge and engagement; and 13 and 14 relate to navigation system and financial circumstances.

Likewise, 5 barriers, lack of knowledge or training about HL and related activities (9/13; 69%) [9,34,37,39,40,41,42,43,46], poor organizational commitment to HL (8/13; 61%) [37,40,41,42,43,45,46,47], the ambiguity of roles among health providers and clinic staff (7/13; 54%) [34,37,40,41,42,43,45], poor interactive and linguistic skills (7/13; 54%) [9,34,37,40,44,45,47], and not having policies, procedures, protocols supporting HL practice (6/13; 46%) [9,34,37,41,43,46], are frequently considered as key barriers that influence quality improvement of OHL practices and its health outcome.

## 4. Discussion

### 4.1. Assessment Tools for Organizational Health Literacy Practice

Although OHL is topical, empirical studies on OHL meeting the needs of patients with low HL are still rare. Similarly, tools designed to improve the quality of OHL are poorly debated in academic databases. Consistently with these gaps, this review highlighted a variety of operational self-assessment tools, quality improvement characteristics of OHL, and its outcome. Similar to HL, OHL seems to be a heterogeneous and complex phenomenon, which has been theoretically deduced from different tools and frameworks [9,37,40,41]. We defined 8 self-assessment tools OHLO, HLHO-10, HL-COM, AHRQ, VHLO, C-CAT, HLE, and Org-HLR that have been evaluated for applicability in health care organizations [20,50].

Although few tools (AHRQ, VHLO, and HLHO-10) address all 10 attributes of OHL, navigation, access, and verbal/written communication are consistently included in all assessment tools. Likewise, all tools address shortages and barriers of HL by enhancing patients’ understanding of health information, reducing thecomplexity of health care, and increasing support for patients with HL at all levels [41]. Health literacy tools, particularly AHRQ, VHLO, and HLHO-10, provide best practices and evidence-based recommendations to enhance OHL capacities [9,37,40,41]; they help build a business case and include facilitating quality improvement processes for OHL. The studies included in the systematic review shows that all tools are potentially validated measurement tools for screening and evaluating OHL, and are implemented according to their quality improvement characteristics and HL dimensions [28].

Carmela Annarumma proposed that AHRQ aims to evaluate the ability of pharmacies to meet patients’ needs and information. In particular, this tool assesses the HL-related readiness of the pharmacies from three different perspectives: staff, patient, and environment. The focus of this self-report survey is the sensitivity of pharmacy staff to HL-related issues, the friendliness of interpersonal communication between patients and pharmacy staff, and the accessibility of printed materials [45].

In 2010, Wynia and Osborn suggested a multi-faceted self-assessment tool for assessing the intensity of OHL, called C-CAT. It includes 9 attributes, that are assessed both from the point of view of patients and from the perspective of health providers and professionals, to globally check the prospective actions to improve OHL and the friendliness of health care organizations [42,51,52]. The C-CAT benchmark has been focused on five out of the nine dimensions of OHL: the engagement of health providers and professionals in the promotion of OHL, the organizational commitment toward the improvement of HL, the methods, and tools used to assess HL, the involvement of the specific communities (included patients, informal caregivers, educational institutions, and municipalities) to improve HL, and the assessment of the initiatives directed at the improvement of OHL [21,28]. This tool is employed to improve HL in the health care context based on two categories. The first one is the formal initiative that includes explicit activities to support the patient in navigating the health services system, and, the second one is the informal initiative that includes voluntary actions of health care professionals to support patients with poor HL [27,39,40,42].

Nancy L. Weaver and Raluca Oana presented that HLE aims to identify facilitating factors and HL-related barriers to healthcare service, navigation, and access from different perspectives (health providers, patient, and environment) and to evaluate ways to better support patients. This tool was organized based on 5 sections (navigation, culture and language, organization policies and practices, and communication) [34].

In V-HLO, Christina C. Wieczorek shows a broader understanding of HL as a coproduction of quality, health promotion, and safety; and “healthy settings.” They also believed that the application of V-HLO goes beyond the scope of medical services [47]. The V-HLO focuses on HL level of health providers, patients, population, and organizations [34,44]. It aims at the sustainability of OHL by focusing on successful disease management, including appropriate access to health care services, the enhancement of organizational capacities and processes to address specific action on HL, patient engagement in the process of care, health promotion at the population level, and disease prevention [34,35,36,37].

Organizational health literacy responsiveness (Org-HLR) is the first empirically developed tool of OHL that considers HL as a determinant of healthcare responsiveness. The Org-HLR framework shows the interconnection between processes and policies, systems, leadership and culture, partnership and community engagement, standards and communication practices, workforce, access to services and programs [46].

Osman Hayran indicated that OHLO and HLHO-10 are an acceptable measurement tool to assess patients’ perceived adequacy of information, and the ability of a health-literate healthcare organization to deal with patients’ HL issues [9].

Nicole Ernstmann indicated that the HL-COM tool is a useful tool for evaluating HL-sensitive communication. This tool could be used to assess communication skills training for health professionals and providers from the patient’s perspective or to measure the characteristics of OHL. However, it is still needs further validation for HL-COM to address its application in different settings [39].

Similarly, the studies included in this review indicate that HLHO-10 and VHLO are the most commonly used tools for assessing OHL capabilities and practices. This may be due to the common usage of these tools and their association with the ten attributes of OHL from different perspectives (patient, health provider, and organization) [34,35,36,37]. All tools include recommendations for improving the care process, such as self-management, discharge and admission, and medication coordination. However, HLHO-10 and VHLO recommend that patient engagement in OHL assessment, service redesign, development of educational materials, and quality improvement efforts [34,44]. The workforce has also included promoting active participation both in the development of a health-literate environment and in the process of care. These tools emphasize the critical role of leadership and culture on the integration of HL in an organization’s mission, and strategic planning, and vision [27,39,40].

### 4.2. Quality Improvement Characteristics of OHL Practice

One of the main objectives of this review was to identify quality improvement characteristics of OHL practices. As shown in the systematic review, quality improvement characteristics are included in 4 main areas: (1) HL is an organizational priority, (2) communication practices and standards, (3) verbal/written communication skills, and (4) patient knowledge and engagement, are commonly recommended as the main quality improvement characteristics to promote sustain HL activities and health outcome. An organization with HL priority includes a range of changes in operational planning and strategic, structures, goals, and processes facilitating quality improvements in the design of services and programs, navigation, policies, protocols, preparation of workforce, and procedures in OHL [53,54]. The organization makes sure that HL is integrated into all relevant planning and programs to deliver health-literate care [29]. These improvements also consist of both targeted and systemic improvements in specific procedures improvements in specific structures and processes (e.g., development of personal care procedures, referrals service, and use of patient portals) [9,55].

Communication practices include a series of improvements in promoting productive interaction [56] and existing health information forms or materials [57]. It also includes of strategies to improve written, spoken, comprehension of health information, and other forms of communication with patients and family (e.g., cross-cultural communication) [29,57]. Finally, patient knowledge and engagement include strategies to promote patient involvement in the health care system and process (e.g., engaging patients in health decision-making and establishing self-care goals) [56,58] and self-management ability and skills [59,60]. Based on this, we recommend that practices related to improving OHL, use relevant strategies in these quality improvement characteristics can decrease the patient’s offer and demands and enhance patient’ self-management [29,61].

Some studies in this review described in OHL policies and methods are likely to improve access to services and programs, continuity and integration of HL practices and care, navigation quality, workforce, leadership and commitment, and HL friendliness of environments in specific settings, but they are less likely to provide relevant and effective methods and policies to improve health care system change [18,59,60]. Similarly, the research included in this review shows that pharmacies are an important unit to meet the special information needs of patients with low HL, and therefore can compensate for the tendency of health service providers to overestimate personal HL skills [45]. Pharmacists are the main mentors for patients since they provide them with information and advice that enhances patient’s ability to successfully function within the health care system. Likewise, HL should be considered as a driver to enhance awareness of the pharmacy’s role to raise the appropriateness and the effectiveness of the health organization service [34,44]. This is completely true when patients face multiple diseases and different medication treatments; they need relevant timely health information materials to improve their skills in to navigate the health care service system [8,52]. However, the findings of this paper revealed that the organizational commitment of pharmacies to the improvement of organizational HL is still poor [45]. In summary, this study recommended that less expenditure, a higher level of self-care, and better health outcomes, could be achieved by meaningful efforts in advancing the organizational health literacy of pharmacies [45].

### 4.3. Barriers and Facilitators of OHL

This review identified lack of knowledge or training about HL, poor organizational commitment to HL, ambiguity of roles between health providers and clinic staff, poor interactive and linguistic skills, not having policies, and procedures, protocols supporting HL practice as the most common barriers for sustainability and integration of OHL [37,40,41,42,43,45,46,47]. The research in this review shows that dealing with these barriers, patients do not have the appropriate abilities and skills to cope with the health system, and therefore have a negative impact on the appropriateness and quality of health care [9,34,37,39,40,41,42,43,44,45,46,47]. Despite HL is acknowledged as a key determinant of quality improvement in the health care system and the enhancement of OHL is considered as the main ingredient in the health instruction for the transition from care to well-being, it seems that the organizational sensitivity and commitment to HL issues is still poor because HL is not included in the corporate identity of health organizations and their strategic programs and planning [4,46].

It seems that the transition to a healthcare organization with comprehensive health knowledge is still ongoing. This can be a complicated process and takes many years. Most healthcare organizations require radical, simultaneous, and multiple changes to transform into a health-literate organizations [9,59]. In particular, healthcare organizations with poor commitment and leadership to HL, therefore, their efforts to improve OHL through agreements, policies and health education interventions are insufficient [4,46]. Due to the lack of knowledge and training about HL and its effects on the sustainability of the health system and health outcomes, HL is not usually integrated into organizations’ vision and strategic planning [60,62,63]. If there is no commitment and support to improve the role of HL and communication, staff and health service providers may not be able to implement HL’s practices and skills due to a lack of familiarity with the concept of HL [35,58]. Key factors may promote the transition to a comprehensive health-literate healthcare organization, including ownership of changes, subcultural diversity within healthcare organizations and systems, and supporting external stakeholders, leadership, and professional allegiance [60,62,63].

### 4.4. Limitation

There are limitations to this review. First, although we have made great efforts to use several search strategies and select eligible studies, it is possible that some of the studies be inadvertently lost. Second, due to cost and time limitations, only English-language texts were examined in this review, which in turn led to the loss of some studies. Third, the lack of relevant keywords in the search list may also result in the deletion of some related reports. Fourth, the generalizability of our findings could be limited because the majority of the publication in this review from the US.

## 5. Conclusions

OHL is emerged to help improve patient’s engagement in the health care process, the navigation system of healthcare, and communication tools and skills. To support this task, several operational tools and frameworks have been (and continue to be) developed and provide best practices and recommendations to implement HL issues in the healthcare organizations. Our findings indicated the fact that all HL tools and frameworks included in this review are overwhelmingly positive because these tools, particularly AHRQ, VHLO, and HLHO-10, provide best practices and evidence-based recommendations to enhance the capacities of healthcare organizations.

It seems that shifting to effective health-literate healthcare organizations may be a complicated process, and health care reform must consider broader health determinants. Based on this, the authors argue that a healthcare system level effort is needed a systemic approach to enhance communication practices and standards, verbal/written communication skills, and patient knowledge and engagement, and that there is a need for the healthcare system to consider HL as an organizational priority, that is, be responsive. This approach will likely improve the healthcare system’s strategy, vision, human resource management, mission, and operation.

In this study, lack of knowledge or training about HL, poor organizational commitment to HL, ambiguity of roles among health providers and clinic staff, poor interactive and linguistic skills, not having policies, and procedures, protocols supporting HL practice are the most common challenges perceived as barriers for sustainability and integration of OHL. This suggests that, as described in AHRQ, VHLO, and HLHO-10, intervention plans should be implemented to correct these barriers to improve the quality of care, health outcomes, and stakeholder capabilities.

## Figures and Tables

**Figure 1 ijerph-17-07544-f001:**
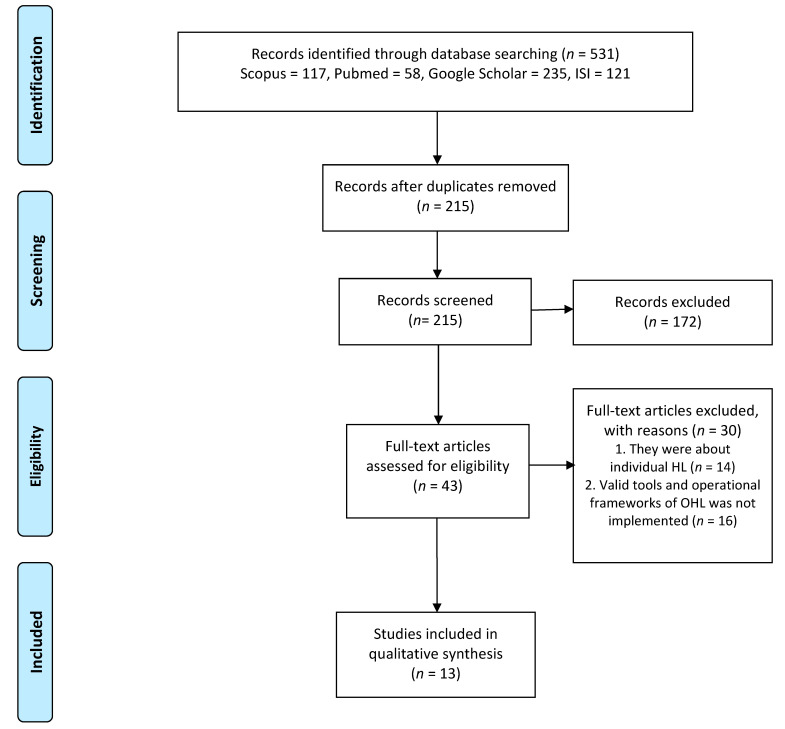
Preferred reporting items for systematic reviews (PRISMA) flow diagram.

**Table 1 ijerph-17-07544-t001:** Inclusion/Exclusion Criteria.

	Inclusion Criteria	Exclusion Criteria
Publication Type	Article, book, guideline, fact sheet, fact book.	There is no limit
Setting	Any type of healthcare setting	
Dissemination Type	Scientific publications in full-text format, which was published in indexed scientific journals	Full-text articles were not published
Outcomes	Publication discusses at least one of the following issues: Studies that have examined OHL in terms of tool creation, development, and validation, or completed OHL standard tools in the environment, or studied OHL formation, implementation, and evaluation, and examined OHL barriers, facilitators, and outcomes in practice.	Articles not meeting the above criteria were excluded.
Language	English	Articles written in languages other than English
Time	From inception until November 2019	

**Table 2 ijerph-17-07544-t002:** Characteristic Included Study.

	Authors (Year)	Country	Study Design	Aims	Health Sector	Tools	Outcome
**1**	Annarummac (2016) [45]	Italy	Comparative research	Explore the organizational HL	60 unite of Pharmacies	AHRQ	Verbal communication, print information materials, and organizational sensitivity to literacy are the core elements for increasing pharmacies’ ability to support low health literate patients. HL is a driver to enhance awareness of the pharmacy’s role in raise the appropriateness and effectiveness of the health care service system.
**2**	Palumbo R. (2014) [43]	Italy	Preliminary	Perform explorative study on the tools adopt to improve their hosts’ HL	health care organizations	C-CAT	Health care organizations are still far from effectively activating HL pathways. Systemic efforts to acquire awareness of the issue are strongly needed.
**3**	Groene R.O. (2006) [44]	Spain	Preliminary	To pilot an assessment of HL issues within hospital settings.	10 Hospitals	HLE	Confusion and insecurity condition found throughout health care facilities. Navigation quality, written and oral communication are main elements to support low health literate patients.
**4**	Wieczorek C. (2017) [47]	Austria	Qualitative	Develop a self-assessment tool to measure the HL friendliness	628 youth health clinic	V-HLO	The V-HLO has potential in varied settings beyond health care. It is important focus on individual skills and capacities, HL friendliness of environments and the enhancement of individual competencies to improve OHL.
**5**	Trezona A. (2018 [46])	Austria	Workshop, review	Develop the OHL self-assessment tool	4 Primary health Care	Org-HLR	The Org-HLR has potential to assess HL responsiveness strengths, limitations, and quality improvement activities in OHL.
**6**	Henrard G. (2019) [27]	Belgium	Pilot study	To translate and culturally adapt HL questionnaire	5 Hospitals	VHLO	VHLO help hospitals to identify their weaknesses and strengths in terms of HL.
**7**	Hayran O. (2019) [9]	Turkey	Cross-sectional study	To investigate organizational HL in hospitals in Istanbul.	30 hospitals	OHLO/ HLHO-10)	Need improvement on providing access, integration, high-risks and costs. OHL improve interpersonal communication and embedded practices.
**8**	Ernstmann N. (2017) [39]	Germany	Pilot study	To develop a survey instrument to assess OHL from the patients’ perspective.	453 patients in cancer centers	HL-COM	The HL-COM is a useful tool to assess HL-sensitive communication or communication skills trainings for health professionals from the patient’s perspective. Understanding information/checking comprehension; verbal and written information are the core elements.
**9**	Kowalski C. (2015) [40]	Germany	Pilot study	Developing and validating an HLHO instrument.	51 hospitals	HLHO-10	HLHO-10 is a useful tool to assess the degree to which health care organizations help patients to understand, navigate, and use information and services.
**10**	Weaver N. (2012) [34]	USA	Case Study	To implement and evaluate HL policy action plan	3 health care organizations	HLE	Low awareness of HL within the organization and variation in perceived values of protocols, inter-staff communication, and patient communication are the main gaps.
**11**	Prince L.Y. (2018) [41]	USA	Cross-sectional study	To assess health care practices at an academic health center	Academic Health Centre	HLHO-10	Need for improvements in health care practices to better assist patients with inadequate HL.
**12**	Wynia M.K. (2010) [42]	USA	Cross-sectional study	To explore the relationship between HL status and receiving patient-centered communication	6 hospitals and 7 health clinics	C-CAT	Improving communication quality in OHL help to address the challenges facing patients with limited HL.
**13**	Wray (2019) [37]	USA	Case study	To evaluate a collaborative effort between a health care organization and academic institution to strengthen organizational HL.	Health Care organizations	HLHO-10	Integrated HL practices into clinic systems, garnered leadership and organizational commitment improve interpersonal communication and embedded practices making health education materials more accessible.

AHRQ: Agency for Healthcare Research and Quality, C-CAT: communication climate assessment toolkits, V-HLO: Vienna Health Literate Organization, Org-HLR: organizational health literacy self-assessment, OHLO: organizational health literacy observation, HLHO-10: health literate health care organization 10 item questionnaire, HL-COM: health literacy-sensitivity of communication; HLE: health literacy environment of hospitals and health centers.

**Table 3 ijerph-17-07544-t003:** Characteristics of the Organizational Health Literacy (OHL) Self-Assessment Tools.

Tools	Attributes	Description
**AHRQ**	10: Leadership, integration, workforce, inclusion of the served, HL skills range, communication standards, provide access, media variety, High-risk, costs	(AHRQ) is a 227-page and includes 20 tools or measures with detailed instructions also includes an appendix with 25 additional resources related to addressing HL at the individual level and the system level [21,48].
**HL-COM**	10: Leadership, integration, workforce, inclusion of the served, HL skills range, communication standards, provide access, media variety, high-risk, costs	An instrument measuring OHL from the patients’ perspective. The 16 items rated on a four-point Likert scale ranging from 1 (‘‘I disagree’’) to 4 (‘‘I fully agree’’) [39].
**OHLO**	10: Leadership, integration, workforce, inclusion of the served, HL skills range, communication standards, provide access, media variety, high-risk, costs	In the form, there are questions to evaluate hospitals’ communication systems as well as how user-friendly and health literate their indoor are scores. Ranged from 10 to 40 where high scores indicated a high level of health literacy [9].
**HLHO-10**	10: Leadership, integration, workforce, inclusion of the served, HL skills range, communication standards, provide access, media variety, high-risk, costs	Survey (10 Likert-scale questions), paper or electronic administration, a self-administered survey that rates OHL practices on a scale of 1 to 7 (‘absolutely not’ –‘to a very large extent’) [35,49].
**VHLO**	9: Establish management policy and organizational structures of HL, develop materials and services in participation with relevant stakeholders, qualify staff for health-literate communication with patients, provide a supportive environment, apply HL principles in routine communication with patients, improve the HL of patients and significant others, improve the HL of staff, contribute to HL in the region, share experience and be a role model.	The questionnaire comprises 9 standards, 22 sub-standards and 160 items. The V-HLO focuses on HL of patients, healthcare providers, organizations, and populations.
**Org-HLR**	7: External policy and funding environment, leadership and culture, systems, processes and policies, access to services and programs, community engagement and partnerships communication practices and standards, workforce.	The self-rating tool was divided into seven assessment dimensions, each of which is made up of 1 to 5 sub-dimensions (24 in total), and 135 performance indicators, were drawn from raw data collected during development of the Org-HLR Framework [29,46]. The V-HLO focuses on HL of patients, healthcare providers, organizations, and populations.
**HLE2**	6: Navigation, print communication, oral exchange, technology, policies, protocols.	It is a 164 page and includes measures related to navigation (31 items); print communication (24 items); oral exchange (8 items); availability of patient-facing technologies (18 items); and policies and protocols pertaining to the development and distribution of print materials, using plain language and patients’ native language to communicate and training staff in HL and health communication issues (19 items) [38,44].
**C-CAT**	9: Leadership commitment, information collection, community engagement, workforce development, individual engagement, sociocultural context, language services, health literacy, and performance evaluation.	The C-CAT includes 74 items on the staff survey, the executive leadership survey has 70 items, and the patient survey has 56 items [36].

**Table 4 ijerph-17-07544-t004:** Quality Improvement Characteristics of OHL Practices.

Characteristics	1	2	3	4	5	6	7	8	9	10	11	12	13
1.Access to services and programs							*		*				
2.Leadership and commitment													*
3. Health literacy is an organizational priority	*	*			*	*	*	*		*	*		
4.HL friendliness of environments				*									
5.Continuity and integration of HL practices and care						*							*
6. Communication practices and standards	*		*	*		*	*	*		*		*	*
7.Navigation quality			*						*				
8. Workforce			*										*
9. Verbal/written communication skills	*		*			*	*	*		*		*	*
10. Financial management							*						
11. Patient knowledge and engagement		*		*				*		*			

* It is expressed significant characteristics of OHL in each study.

**Table 5 ijerph-17-07544-t005:** Barriers and Facilitators of OHL.

Barriers and Facilitators	1	2	3	4	5	6	7	8	9	10	11	12	13
1. Poor organizational commitment to HL	*	*		*	*				*		*	*	*
2. Not having enough time and resource constraints	*							*	*				
3. Not having policies, procedures, protocols supporting HL practice		*			*		*			*	*		*
4. The organizations solely focus their attentions on the patient engagement and self-management.		*			*				*	*			
5. Lake of HL friendliness of environments		*			*				*				
6. Poor interactive and linguistic skills (Use of scientific language and abbreviations or inconsistency in the terminology used by staff)	*		*	*			*		*	*			*
7. Ambiguity of roles among health providers and clinic staff to address the health needs and material	*	*							*	*	*	*	*
8. Poor verbal/written communication skills			*	*					*			*	
9. lack of confidence in completing medical forms										*		*	
10. Lack of culture of change and innovation							*	*					
11. Lake of the patient involvement		*							*				
12. Lack of knowledge or training about HL and related activities		*			*		*	*	*	*	*	*	*
13. Poor navigations system (multiple entrances, absence of signs, printed and posted word, visual and physical element)			*	*			*						
14. financial circumstances							*

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
