# Peer review of "Review of Organizational Health Literacy Practice at Health Care Centers: Outcomes, Barriers and Facilitators"

_ijerph, 2020, doi:10.3390/ijerph17207544_

Round 1

Reviewer 1 Report

This systematic review focuses on a very important and newly-emerged concept. Authors did a great job in presenting their methods and the PRISMA diagram is a great addition. However, it will be great if they could report on any assessed bias of individual studies (e.g. publication bias, selective reporting). 

There are several limitations that should be considered when interpreting the findings of this study. However, authors were well-aware of them and presented them in the limitations section.

An extensive editing of the English language is required.  

Author Response

We would like to thank you for careful and thorough reading of this manuscript and for the thoughtful comments and constructive suggestions, which help to improve the quality of this manuscript. We tried to answer to all valuable comments/suggestions/queries and all answers to comments are rewritten within the document by italic format.

Comment 1: This systematic review focuses on a very important and newly-emerged concept. Authors did a great job in presenting their methods and the PRISMA diagram is a great addition. However, it will be great if they could report on any assessed bias of individual studies (e.g. publication bias, selective reporting).

Reply 1: All doubt and disagreements regarding data extraction were resolved by discussion among authors and the independent dual rating was also performed based on the Cochrane diagnostic test accuracy Reviews to estimate the quality of included studies. The quality of the included studies was acceptable with an average score of 11.5. It ranged from 8.4 to 14. This is high quality consistent with QUADAS tools for systematic review studies. In case of selective report, details of the exclusion and inclusion criteria were provided in Table 1. Also, to increase the accuracy of the study, all scientific publications in full-text format, which was published in the indexed scientific database included in this review. However, original research studies were given preference because they provide details about methods

Comment 2: There are several limitations that should be considered when interpreting the findings of this study. However, authors were well-aware of them and presented them in the limitations section.

Reply 2: limitations was updated as follow: There are limitations to this review. First, although we have made great efforts to use several search strategies and select eligible studies, it is possible that some of the studies be inadvertently lost. Second, due to cost and time limitations, only English-language texts were examined in this review, which in turn led to the loss of some studies. Third, the lack of relevant keywords in search lists may also lead to the deletion of some related reports. Fourth, the generalizability of our findings could be limited because the majority of the publication included HL tools in this review from the US.

Comment 3: An extensive editing of the English language is required. 

This manuscript was edited by native English language and all changes were rewritten by BLUE color.

Reviewer 2 Report

The study presents an interesting research supported by an exhaustive literature review. However, the contribution of authors associated with OHL is not evident.

The authors emphasize the importance of the OHL through bibliographic research, but the analysis lacks a more evident segmentation. It is  scientifically correct to include the different health institutions (hospitals, pharmacies, etc.)? Or should they correspond to different analysis segments?

It seems interesting to me to carry out a study on OHL and to compare it with other existing ones.

Author Response

We would like to thank you for careful and thorough reading of this manuscript and for the thoughtful comments and constructive suggestions, which help to improve the quality of this manuscript. We tried to answer to all valuable comments/suggestions/queries and all answers to comments are highlighted within the document by Yellow format.

Comment 1. The study presents an interesting research supported by an exhaustive literature review. However, the contribution of authors associated with OHL is not evident.

Reply 2: based on this comment, relevant information to clarify the contribution of authors with OHL and their motivation was explained to introduction section as follow:

Several health strategies are designed to improve health behavior, self-care abilities, and quality of life, but adherence and participation are related to social health determinants[8, 9]. A large study showed that HL is a main modifying factor for health outcomes because is associated with social health determinants[10, 11]. Although, organizational initiatives responding to HL may be a practical approach to improve healthcare systems and health-promoting hospitals, using experiences from the quality movement in these systems remains less responsive than needed to the concept of limited HL.

Despite the recent research on designs and structures of OHL, the effectiveness and use of these attributes have not been reviewed systematically to understand the outcomes and implications of transformation to the health-literate organization [1-4]. The scientific studies have focused most of its attention on patient treatment; in contrast, the role of OHL in empowering patient’s self- management support and communication has been overlooked[5]. Likewise, limited work has been done to identify barriers and facilitators that influence quality improvement in OHL[6, 7]. Absent such information, health systems may be unable to evaluate whether OHL-related attributes and guidelines have been implemented effectively or to identify the characteristics and outcomes of its environment most in need of improvement[8].  Therefore, in this review, we aim to address these specific gaps in knowledge by focusing on the following: (1) the evidence for the effectiveness of OHL practices and outcome at a healthcare center by using the criteria found in the 10 attributes, and (2) the facilitators and barriers that influence the implementing OHL.

We used a systematic review as a tool to conduct a comprehensive assessment of OHL improvement. This review provides insights into the implementation of OHL that are less visible to health providers and patients, and identify facilitators and barriers that influence the quality of organization and delivery of healthcare systems. These findings have relevance for health providers, professionals, and researchers working to improve OHL and evaluating the quality of care.

Comment 2: The authors emphasize the importance of the OHL through bibliographic research, but the analysis lacks a more evident segmentation. It is scientifically correct to include the different health institutions (hospitals, pharmacies, etc.)? Or should they correspond to different analysis segments?

Reply 2: based on this valuable comment all parts of results and discussion was re-written and further information were explained. 

Reviewer 3 Report

Dear Authors

I carefully evaluated the study, finding it not well written and presented. First of all, a revision of the language by a native English language is strongly recommended. The article needs strong improvements before considering it as suitable for publication.

Line 1: type of article: it is a review

Abstract: provide a brief definition of OHL

Introduction: Authors should better explain the literature gap they want to bridge with the present study. Study motivation should also be presented in a more explicit way. At this stage, study motivation appears very weak.

The type of review must be presented early in the paper.

Methods: why no study design limitation was considered?

Figure 1 “they were the concept of health literacy organizations”. This sentence is not clear. Why these papers have been excluded?

Table 2. Check the term “Italia”

Authors should motivate the choice to focus on the countries that are working on OHL. Why the Authors considered this aspect as worthy of attention?

I can not find any significant element of interest in the paragraph 3.2. It is not clear the reason why it has received so attention.

In the methods section, Authors stated: “Independent dual rating was also performed to estimate quality of included studies”. I can not find any result about this rating. Add the results about the quality rating you performed.

Discussion. The results of the present review should be discussed in a more organic way. As presented by the Authors, they seem as separated part of the same problem, without a clear link between them. I suggest to improve the discussion section to provide some clear take-home messages and to provide some useful information for organizations and stakeholders who want to manage this relevant issue.

Conclusions are very confusing. Authors should state clearly what is the contribution of this review on the knowledge about the theme of OHL. Conclusion section has to be substantially rewritten. Authors must go further the simple description of the included study. The overall interpretations, practical implications and theoretical contribution are very lacking. At this stage, the scientific contribution of the present study is questionable.

Author Response

Comment 1: I carefully evaluated the study, finding it not well written and presented. First of all, a revision of the language by a native English language is strongly recommended. The article needs strong improvements before considering it as suitable for publication.

Reply 1: We would like to thank you for careful and thorough reading of this manuscript and for the thoughtful comments and constructive suggestions, which help to improve the quality of this manuscript. We tried to answer to all valuable comments/suggestions/queries and all answers to comments are rewritten within the document by yellow highlights. Also, this manuscript was edited by native English language and all changes were rewritten by BLUE color.

Comment2: Line 1: type of article: it is a review

Reply 2: Done

Comment 3: Abstract: provide a brief definition of OHL

Reply3: it was added as follow: the term Organizational health literacy (OHL) describes a healthcare organization that use strategies to make it easier for patients to engage in the health care process, navigate the health care system, understand health information, and manage their health

Comment 4: Introduction: Authors should better explain the literature gap they want to bridge with the present study. Study motivation should also be presented in a more explicit way. At this stage, study motivation appears very weak.

Reply 4: based on this comment: further information related to literature gap and study motivation were added.

Comment 5: The type of review must be presented early in the paper.

Reply 5: based on this comment. The type of review was presented in early in abstract as follow:  In this study, a systematic review of the literature was used to understand the evidence for the effectiveness of OHL and its health outcome, and the facilitators and barriers that influence the implementing OHL

Comment 6: Methods: why no study design limitation was considered?

Reply 6: No study design, and country restrictions were imposed. To increase the accuracy of the study, all scientific publications in full-text format, which was published in indexed scientific database included in this review. However, original research studies were given preference because they provide details about methods

Comment 7: Figure 1 “they were the concept of health literacy organizations”. This sentence is not clear. Why these papers have been excluded?

Reply 7: It was corrected in Figure 1. These paper were excluded because valid theories and operational frameworks of OHL was not implemented.

Comment 8:  In the methods section, Authors stated: “Independent dual rating was also performed to estimate quality of included studies”. I can not find any result about this rating. Add the results about the quality rating you performed.

Reply 8: All doubt and disagreements regarding data extraction were resolved by discussion among authors and the independent dual rating was also performed based on the Cochrane diagnostic test accuracy Reviews to estimate the quality of included studies. The quality of the included studies was acceptable with an average score of 11.5. It ranged from 8.4 to 14. This is high quality consistent with QUADAS tools for systematic review studies.

Comment 9: Table 2. Check the term “Italia”

Reply 9: it was corrected

Comment 10: I can not find any significant element of interest in the paragraph 3.2. It is not clear the reason why it has received so attention.

Reply 10: it was removed from this manuscript.

Comment 11: Discussion. The results of the present review should be discussed in a more organic way. As presented by the Authors, they seem as separated part of the same problem, without a clear link between them. I suggest to improve the discussion section to provide some clear take-home messages and to provide some useful information for organizations and stakeholders who want to manage this relevant issue.

Reply 11: based on this valuable comment all parts of results and discussion was re-written and further information were explained. 

Comment 12: Conclusions are very confusing. Authors should state clearly what is the contribution of this review on the knowledge about the theme of OHL. Conclusion section has to be substantially rewritten. Authors must go further the simple description of the included study. The overall interpretations, practical implications and theoretical contribution are very lacking. At this stage, the scientific contribution of the present study is questionable.

Reply 12: based on this valuable comment all parts of conclusion was rewritten and further information were explained as follow:

OHL is emerged to help improve patient’s engagement in the health care process, the navigation system of healthcare, and communication tools and skills. To support this task, several operational tools and frameworks have been (and continue to be) developed and provide best practices and recommendations to implement HL issues in the healthcare organizations. Our findings indicated the fact that all HL tools and frameworks included in this review are overwhelmingly positive because these tools, particularly AHRQ, VHLO, and HLHO-10, provide best practices and evidence-based recommendations to enhance the capacities of healthcare organizations.

It is seeming that shifting to effective health-literate healthcare organizations would likely be a complex process and healthcare changing has to consider the wider determinates of health. Based on this, the authors argue that a healthcare system level effort is needed a systemic approach to enhance communication practices and standards, verbal/written communication skills, and patient knowledge and engagement, and that there is a need for the healthcare system to consider HL as an organizational priority, that is, be responsive. This approach will likely improve the healthcare system’s strategy, vision, human resources management, mission, and operation.

In this study, lack of knowledge or training about HL, poor organizational commitment to HL, ambiguity of roles among health providers and clinic staff, poor interactive and linguistic skills, not having policies, and procedures, protocols supporting HL practice are the most common challenges perceived as barriers for sustainability and integration of OHL. This suggests that interventions programs should be implemented to remediate these barriers to improve quality of care quality, health outcomes, and stakeholder empowerment as explained in AHRQ, VHLO, and HLHO-10.

Round 2

Reviewer 3 Report

Dear Authors

All th concerns about the paper have been solved.

Best regards

Author Response

more thanks 
